# Positive Parenting and Sociodemographic Factors Related to the Development of Chilean Children Born to Adolescent Mothers

**DOI:** 10.3390/children10111778

**Published:** 2023-11-02

**Authors:** Laura Léniz-Maturana, Rosa Vilaseca, David Leiva, Rodrigo Gallardo-Rodríguez

**Affiliations:** 1Department of Cognition, Development and Educational Psychology, University of Barcelona, 08007 Barcelona, Spain; llenizma37@alumnes.ub.edu; 2Department of Social Psychology and Quantitative Psychology, University of Barcelona, 08007 Barcelona, Spain; dleivaur@ub.edu; 3Department of Sport Science and Physical Conditioning, Faculty of Education, Universidad Católica de la Santísima Concepción, Concepción 4070129, Chile; rgallardo@ucsc.cl

**Keywords:** adolescent mothers, positive parenting, child development, family sociodemographic factors

## Abstract

The lack of economic resources has a negative effect on the maternal role of younger mothers. In Chile, the majority of adolescent pregnancies occur in socially and economically vulnerable contexts. The current study aimed to examine the relationship between demographic variables within the family context and parenting behaviors among Chilean adolescent mothers (including affection, responsiveness, encouragement, and teaching). These factors were correlated with communication, problem-solving abilities, and personal–social development in typically developing infants. The study included a sample of 79 Chilean adolescent mother–child dyads with children aged 10 to 24 months. Communication, problem-solving, and personal–social development were assessed using the Ages and Stages Questionnaire-3, along with a demographic information questionnaire. The parenting behaviors mentioned above were observed using the Spanish version of Parenting Interactions with Children: Checklist of Observations Linked to Outcomes. The findings indicated that mothers in employment and those who had not dropped out of school had children with better problem-solving skills. Additionally, children residing with their fathers and female children performed better in communication, problem-solving, and personal–social development. Maternal responsiveness was associated with communication and problem-solving, while maternal encouragement was linked to improved problem-solving skills. Maternal teaching was connected to communication, problem-solving, and personal–social development. The study emphasized the significance of parenting and sociodemographic factors among adolescent mothers and their influence on their children’s development.

## 1. Introduction

Parenting is a broad and diverse concept that involves different aspects of childrearing, including support for children’s needs, parental styles, provision of food and care, teaching and developmental guidance, and affectionate communication [1]. Following Roggman et al. [2], in this paper, “parenting” refers to the characteristics of high-quality parent–child dyadic interactions that occur face-to-face and promote child development. The relationship between parenting and child development has been studied in high-risk families, specifically in families from socially and economically vulnerable contexts [3,4,5]. The research has demonstrated that low economic income negatively impacts the family context; home learning environments for children tend to be less stimulating in this situation [6], and the ability of younger mothers to carry out their maternal role may be directly affected [7]. Scientific evidence has consistently demonstrated the harmful impact of socioeconomic risk on adolescent mothers and, as a result, on their children’s development [8]. According to the World Health Organization [9], adolescent motherhood is more prevalent among women from low-income country. Latin America has the second-highest adolescent birth rate in the world, with a higher prevalence in lower-income countries [10]. In Chile, pregnancies among adolescents from economically and socially vulnerable backgrounds are frequent [11], particularly in the Biobío region, which has some of the highest poverty rates in the country [12] and a higher than average rate of adolescent mothers living in circumstances that may violate their legal rights [13]. The present study set out to explore the relationship between sociodemographic factors and positive parenting in Chilean adolescent mothers and their children.

### 1.1. Chilean Adolescent Mothers’ Sociodemographic Context and Children’s Developmental Outcomes

The children of families at high social and economic risk are significantly more likely to present delayed cognitive and linguistic development [14], possibly due to the adverse effects of reduced stimulation of cognitive skills and suboptimal parenting practices [15]. Previous research has suggested that children born to adolescent mothers tend to exhibit slower cognitive, linguistic, and social development compared to those born to older mothers, possibly due to the highly vulnerable context in which they are raised [16,17,18,19,20]. Moreover, low socioeconomic status and poverty can interfere with cognitive development by limiting the possibilities for proper childrearing and stimulation [8].

In Chile, maternal skills are positively correlated with higher economic income, educational level, and older maternal age [21]. Chilean adolescent mothers are more likely to drop out of school and therefore have lower educational levels [22]; this increases the risk of future unemployment and ultimately perpetuates the cycle of poverty [23]. The consequences of unemployment for a mother who became pregnant during adolescence are important. In a study by Reynolds et al. [24], Chilean children whose mothers were working during their early years demonstrated better performance in cognitive, linguistic, and socio-emotional stimulation. This may be attributed to the mothers’ employment status, as it was associated with a more enriching learning environment able to promote their children’s development [25]. The employment and educational status and the high-risk and vulnerable context of adolescent mothers highlight the need for parenting support to optimize their children’s development [26]. The study by Luttges et al. [27] found that Chilean adolescent mothers required more guidance on parenting strategies, probably because they often lacked the maternal skills necessary to face adverse situations [28,29]. It is important to note that extended family support for adolescent mothers can contribute to maternal satisfaction [30]. In fact, child development may be less affected if grandparents offer protection against the adverse effects of fatherlessness [31]. Furthermore, childcare support for adolescent mothers encourages them to be more affectionate with their babies [32]. Therefore, it is necessary to explore the impact of support in caring for babies when these mothers show affection during interactions on their child’s development.

While social support for adolescent mothers is essential, research suggests that the absence of fathers is one of the primary challenges young mothers face and can significantly impact their children’s development in terms of academic performance, enhanced reading skills, and fewer behavioral problems in later years [33]. Moreover, fathers’ early involvement significantly influences children’s neurobiological development, emotional and behavioral regulation, and their self-concept and perceptions of others [34]. In the Chilean context, it has been observed that the absence of adolescent fathers in the home may contribute to an unstable relationship characterized by a low level of commitment, potentially resulting in relationship breakdowns and subsequent loss of contact and bonding with their children [35].

Another aspect to consider with regard to the support in the care of adolescent mothers’ children is that Chile offers preschool centers designed for adolescent mothers’ children, which are free of charge [36]. Their primary goal is to assist young mothers and fathers in ensuring their children receive the necessary care and stimulation, and enabling adolescents to continue their education, because early motherhood often leads to high school dropout rates among young women. In Chile, efforts have been made to improve the quality of early childhood education to provide children from impoverished backgrounds with the same opportunities as those from higher-income families. [37]. Therefore, it is essential to analyze the effects on child development of attending preschool when their mothers do not receive support in their care.

It has been shown that parenting depends on various sociodemographic variables, such as parent’s age, employment, and educational level [38]. This study investigates the association between these factors in the development of children of adolescent mothers.

### 1.2. Positive Parenting in Adolescent Mothers and Children’s Developmental Outcomes

Studies increasingly recognize the value of parenting quality in child developmental outcomes [39,40,41]. However, adolescent mothers are characterized by having a level of negative involvement, resulting in less positive interaction and increased withdrawal from their children [42]. The family context can influence the mother’s emotional state during these interactions [43]. Therefore, the significance of positive parenting in adolescent mothers and its impact on their children’s developmental outcomes is crucial due to its effect on children’s language skills, cognitive abilities, and socioemotional development [44,45,46]. These interactions promote the expansion of vocabulary and early literacy, fostering expressive language development, executive functioning, academic achievement, memory, and problem-solving skills [47,48,49,50,51]. Furthermore, positive parenting practices predict higher levels of child social self-efficacy in Latino mothers [52], covering various parenting behaviors in domains such as affection, responsiveness, encouragement, and cognitive stimulation or teaching, all of which are closely associated with children’s developmental outcomes [2,53,54].

Affection or emotional warmth, which refers to the behaviors of warmth and fondness towards the child [2], is associated with fewer problem behaviors in infants [55] and higher social skills and communication abilities in early childhood [56]. It predicts lower levels of child externalizing problems [57] and higher levels of executive function in preschool and post-kindergarten [58], which are associated with cognitive development.

Responsiveness behaviors are contingent responses to the child’s initiatives and behaviors. This parenting behavior promotes higher pre-academic abilities, such as reading comprehension, applied math problems, and receptive vocabulary [59], and predicts higher cognitive development in infants [60] related to better memory and, in turn, better social development and self-regulation [61]. Both affection and responsiveness in adolescent mothers can be affected by socioeconomic risk, influencing children’s cognitive development [8] and infant security [62].

Encouragement, understood as promoting autonomy in children, positively impacts independence, security, and children’s language, cognitive, and socioemotional development [63]. However, extreme intrusiveness can trigger or increase children’s separation anxiety [64]. Children born to adolescent mothers may present lower levels of expressive and receptive language due to this high intrusiveness, rather than encouragement, and low verbal stimulation [20].

Teaching or cognitive stimulation, through joint parent–child activities and conversation enhances cognitive skills, receptive and expressive communication, and prosocial behavior [40,65,66]. In contrast, low verbal stimulation by adolescent mothers can hinder their child’s development.

Another interesting aspect to consider is whether parenting styles may differ depending on the gender of the children. In general, few differences have been found between boys and girls. Autonomy support strategies appear to be encouraged more in boys than in girls [67]. A recent systematic review showed differences in parenting with respect to the language directed to children according to their gender, socialization strategies, play, and the provision of different materials and toys, which clearly influenced child development [68]. Other studies showed that mothers are more affectionate with girls when their distress levels are lower, which is associated with living with the biological father [69,70,71].

The present study aims to provide insights that enable the implementation of intervention practices to enhance the quality of interaction between adolescent mothers and their children [72,73]. Despite the relevance of investigating adolescent motherhood, most studies of the relationship between parenting and child development have been carried out with adult mothers [74,75,76,77]. Few studies have explored the relationship between parenting in adolescent mothers, family sociodemographic factors, and child development. Various studies have shown that a home environment characterized by positive interactions between parents and children significantly impacts typically developing children [4,78,79]. In the case of adolescent mothers from low socioeconomic backgrounds, achieving positive interactions with their children can be a real challenge.

Based on the evidence reviewed above, this exploratory study aims to: (a) examine the relation between family related demographic variables and parental factors (e.g., school attendance/non-attendance, employment status, support in the care of a child, child’s gender, fathers/grandparents living with their child, and children attending preschool), and language, cognitive, and personal–social development in typically developing young children of Chilean adolescent mothers; and (b) examine the relation between parenting, defined in terms of affection, responsiveness, encouragement, and teaching and cognitive, linguistic and personal–social development in typically developing children of Chilean adolescent mothers at early ages.

## 2. Materials and Methods

### 2.1. Participants

The current study was conducted in seven health centers, one hospital, four preschool learning centers, and one residential home for adolescent mothers (belonging to the National Children’s Service) in Biobío, Chile. Power analyses were run in advance to determine the required sample size for the present study. Analyses based on Fisher’s transformation for the correlation test, and assuming moderate-sized correlations (r = 0.3, α = 0.05, and 1 − β = 0.80), yielded a required sample size of 84 individuals. Additionally, we ran power analyses for linear regressions by assuming models with at least four predictors and moderate effects (k = 4, f2 = 0.15—equivalent to R2 = 0.18, α = 0.05, and 1 − β = 0.80), for which the minimum required sample size was 80 individuals.

The final sample included 79 dyads of mothers and children who met the eligibility criteria (low-income mothers who became pregnant at 19 years old or younger, average household income less than or equal to USD 678.49 per month for mid–low income, and USD 391.16 per month for low income, according to the criteria of the Chilean Association of Market Researchers 2019 [80], and children of typical development. Within this group of mothers, 62% had completed high school, 29% had only completed primary school, and fewer than half had not completed primary school (9%, n = 7); 14% had dropped out of school due to maternity.

Adolescent mothers aged between 15 and 21 years (M = 19.1, SD = 1.7) and children aged between 10 and 24 months (M = 15.5, SD = 4.2) were visited at their homes. They provided voluntary, informed consent for their participation and use of data related to their family’s sociodemographic, parenting, and child developmental characteristics when older than 18 years old; in the case of mothers younger than 18 years old, their legal guardians signed informed consent. The characteristics of the children and their mothers that were relevant to the study are presented in Table 1.

### 2.2. Procedure

Data procedures were collected via questionnaires and observations. Firstly, the coordinators of the health centers, hospitals, preschool learning centers, and residential homes for adolescent mothers provided a database with telephone numbers and addresses to contact the families directly. Families who agreed to participate were visited at home and informed about the study. A sociodemographic questionnaire was read and explained to adolescent mothers, clarifying the possible answers. Subsequently, mothers were asked to engage in a video-recorded 10 min play session with their infants at home, with the following instruction translated into Spanish: “Interact and play with your children as you typically do.” Mother–child dyads were observed during 10 min of free play using toys provided in three bags that included books, toys for pretend play, and age-appropriate manipulative toys, respectively. The “Three Bag task” provides a structured space with flexibility for the mother to guide the interaction and to use the toys as they wish while playing with their children [81]. PICCOLO [2] was then used to score parent–child interactions in these 10 min video recordings. Finally, the Ages and Stages questionnaire-3 [82] was administered to check whether the child performed the behavior indicated in the items of the areas of communication, problem-solving, and personal–social with the following instruction: “We will read each question carefully and check whether your baby does the activity, does the activity sometimes, or not yet.” Interviewers and mothers completed the ASQ-3 in around 10–15 min. According to Small et al. [83], in low-income countries in Latin America, this instrument is usually read by a trained evaluator who helps parents record and interpret their child’s responses when they are evaluated. Mothers completed ASQ-3, but all items were read and verified whether the child did the activity indicated in the questionnaire by a member of the research team due to difficulties in reading presented by the adolescent mothers.

### 2.3. Measures

An ad hoc sociodemographic questionnaire was used to record the mother’s age, educational level, employment status, and whether she was receiving support or help in the care of her child from the child’s grandparents when the mother was studying or working at home and in parenting tasks such as feeding, toileting, and putting the baby down to sleep. The same questionnaire was used to record the child’s age, gender, relatives who resided with them, and whether they attended a preschool learning center taking into account whether or not they regarded it as an option for improving the child’s development, and support in the care of their child.

The Parenting Interactions with Children: Checklist of Observations Linked to Outcomes (PICCOLO, [2]) is an observational instrument that assesses interactions between parents and children aged 10 to 47 months. It includes 29 observable parenting behaviors that reflect parent interaction behaviors, which are rated for frequency as 0 (absent, no behavior observed), 1 (bare, minor, or emergent behavior), and 2 (clear, definite, and behavior frequent). These are grouped into four domains: (a) Affection (7 items), which involves physical and verbal expression of affection; (b) Responsiveness (7 items), which includes reacting sensitively to the child’s cues and interests; (c) Encouragement (7 items), which considers parents’ support for efforts to promote children’s autonomy; and (d) Teaching (8 items), which includes cognitive and linguistic stimulation, joint attention and play. The instrument generates a score for each domain between 0 to 14 (0 to 16 for the teaching domain) and a total score between 0 and 58 (adding all the items). The original PICCOLO reliability is good; the analysis of Cronbach α for the total instrument was 0.91 (0.78 for affection, 0.75 for responsiveness, 0.77 for encouragement, and 0.80 for teaching), and the instrument had good results for construct and predictive validity [6]. In this study, the Chilean adaptation of PICCOLO [84] was applied, and the reliability (Cronbach α) for the total instrument was 0.88 (0.76 for affection, 0.85 for responsiveness, 0.75 for encouragement, and 0.65 for teaching). For this sample comprised of Chilean adolescent mothers (n = 79), the reliability (Cronbach α) for the total instrument was 0.87 (0.64 for affection, 0.70 for responsiveness, 0.69 for encouragement, and 0.62 for teaching). Inter-rater reliability for the current study was calculated from 20% of observations, yielding an inter-rater agreement estimate of 0.85 for total scores used in the current analyses. The training of the raters involved the following steps: the second author, whom the authors of the original PICCOLO trained, trained the first author for this study, who read about the content and purpose of the measure (during an approximately 3 h session) during a university course attended by the first author that the second author dictated. In this course, the first author scored the Spanish version of the PICCOLO when she watched four video recordings (3 h). The first author was considered to have satisfactorily completed her training when the percentage of agreement between evaluators was equal to or higher than 80%. Both authors viewed and discussed 20 of the 79 video recordings for this study and coded them to establish reliability.

The Spanish version of the Ages and Stages Questionnaire Third Edition (ASQ-3), developed by Squires et al. [82], is a caregiver report tool used to assess children’s developmental progress from birth to 60 months. This questionnaire includes six items that pertain to the areas of communication (both comprehensive and expressive language), problem-solving (which includes cognitive skills in terms of learning and play), and personal–social development (related to the child’s self-help abilities and social interactions). For each item, respondents select one of three possible responses coded as “10” (yes), “5” (sometimes), or “0” (not yet), based on whether the child could perform a specific task. The overall score is derived by summing these responses, with higher scores indicating higher levels of development in children. To ensure consistency and avoid variations in cutoff scores based on age ranges defined in ASQ-3, the developmental scores of children in this sample were transformed into Z scores using a standard metric. The ASQ is recognized as a valid and reliable instrument for assessing child development, with a reported Cronbach α coefficient of 0.94 for overall reliability [85]. It should be noted that ASQ-3 was validated in Chile by Armijo et al. [86] and is considered an effective screening tool. In the present sample, reliability was confirmed with a Cronbach α coefficient of 0.73.

### 2.4. Data Analysis

Data were analyzed in several stages. Firstly the bivariate association between parenting behaviors, socio-demographic factors (i.e., whether the mother had dropped out of school, her employment status and whether she received support or help in the care of her child, child’s gender, whether fathers resided with their child, and whether children attended preschool), and the ASQ-3 scores were quantified and tested by means of correlation tests in the case of interval scales and using parametric tests based on comparisons of means in the case of categorical predictors. To complement the statistical analysis, effect sizes were computed using Cohen’s d, eta-squared, and product–moment correlations. In this regard, these analyses were useful for determining initial sets of predictors to be included in the predictive models for all ASQ-3 scores. More specifically, a significance (*p*) lower than 0.05 was used to specify sets of predictors to be used for all responses employed in the current study.

Linear regression models were estimated in order to predict ASQ-3 scores using as exogenous variables all relevant variables concerning parenting behaviors as well as socio-demographic factors employed as potential predictors in the modeling procedure. In this regard, the routine was based upon a feasible solutions algorithm [87], an intensive computing method that allows researchers to find an optimal solution amongst multiple possible solutions (i.e., candidate models). These multiple solutions correspond to different subsets of predictors including second-order interaction terms. Specifically, the intensive routine was as follows: (1) In the first step of every iteration evaluated, models were specified to include those predictors that proved to be useful in the bivariate analyses along with a random subset of the remaining predictors with which second-order interactions will be assessed; (2) in the next step, the complexity of the model was trimmed taking into account its predictive capacity; (3) steps 1 and 2 were iterated k times, saving the model obtained in each iteration; (4) the model that yielded the highest number of occasions, and thus having the highest predictive capacity, was kept as the fittest one; 5) in the final step of the modeling routine, some of the terms kept in the fittest model found in step 3 were removed, depending on the goodness-of-fit index. One thousand different initial random subsets of predictors were studied for each response (k = 1,000), allowing us to evaluate the models’ space when searching for an optimal solution while avoiding local solutions. Adjusted R-squared was employed as a fit index during the iterations and the complexity of the final solution obtained by means of the intensive procedure was reduced whenever possible by using Akaike’s Information Criterion (AIC).

All statistical models were assessed in order to improve their specification by adding non-linear terms (i.e., polynomials) as well as detecting possible issues concerning multicollinearity (by inspecting variance inflation factors, VIFs, and considering VIF < 5 as an indicator of the absence of multicollinearity problems). Finally, raw, and standardized effects were estimated for interpretation using the original metrics of the scales used and as dimensional effects, thus allowing comparisons between predictors in terms of their relative importance.

All statistical analyses were carried out using the R environment (version 4.1; R Core Team, 2021). The intensive modeling routine was implemented using the rFSA package [88].

## 3. Results

### 3.1. Sociodemographic Factors and Children’s Developmental Outcomes

The relationship between each factor included in the sociodemographic questionnaire (parenting domains regarding affection, responsiveness, encouragement, and teaching) and children’s developmental outcomes (ASQ-3 Z scores) was analyzed. The results showed in the Table 2 that mothers were more responsive when their children were girls (*t* (77) = −2.19; *p* = 0.02; *d* = 0.49) and when fathers resided with their children (*t* (77) = 2.0; *p* = 0.02; *d* = 0.49). Moreover, mothers’ encouragement scores were higher when mothers did not drop out of school (*t* (77) = −2.05; *p* = 0.02; *d* = 0.67) and when fathers resided with their child (*t* (77) = 1.86; *p* = 0.03; *d* = 0.45). Finally, teaching domain scores were higher when mothers were employed (*t* (77) = 2.73; *p* = 0.02; *d* = 0.62), when their children were girls (*t* (77) = 1.82; *p* = 0.04; *d* = 0.41) and when the father resided with their child (*t* (77) = 1.70; *p* = 0.047; *d* = 0.42). Specifically, teaching domain scores were higher when fathers and grandparents resided with the children [F (2,29.5) = 3.42; *p* = 0.038; η^2^ = 0.08].

Regarding the relationship between children’s development and sociodemographic factors, the results showed associations between mothers dropping out of school (*t* (77) = −2.32; *p* = 0.023 *d* = 0.76), employment status (*t* (77) = 2.07; *p* = 0.042; *d* = 0.60) and children’s problem-solving scores. Children with currently employed mothers who had not dropped out of school showed significantly higher scores in problem-solving development. Additionally, there was a significant association between the fact of fathers residing with their children and scores corresponding to communication (*t* (77 = 2.77; *p* = 0.01; *d* =0.69), problem-solving (*t* (77) = 2.54; *p* = 0.013; *d* = 0.63), and personal–social (*t* (77) = 2.61; *p* = 0.011; *d* = 0.65) development. Specifically, children residing with their fathers scored significantly higher in the abovementioned dimensions.

Likewise, girls performed significantly better than boys in communication (*t* (77) = −2.22; *p* = 0.029; *d* = 0.51), problem-solving (*t* (77) = −2.21; *p* = 0.03; *d* = 0.50), and personal–social (*t* (77) = −2.65; *p* = 0.01; *d* = 0.60) scales.

### 3.2. Parenting Domains and Children’s Developmental Outcomes

Observed average levels for the PICCOLO domains were 9.91 (SD = 2.51; Min = 3, Max = 14) for affection, 10.03 (SD = 2.54; Min = 2, Max = 14) for responsiveness, 8.78 (SD= 2.70; Min = 1, Max = 14) for encouragement, and 8.59 (SD = 3.02; Min = 0, Max = 14) for teaching.

Table 3 shows statistically significant Pearson’s correlation coefficients were found between children’s development and parenting. Firstly, communication development was associated with the mother’s responsiveness and teaching. Likewise, problem-solving was related to the mother’s responsiveness, encouragement, and teaching. Finally, personal–social development was significantly associated with the mother’s teaching.

### 3.3. Predictive Models for Infants’ ASQ-3 Developmental Scores

The results indicate that higher levels in all developmental areas of the ASQ-3, that is, communication (β = −0.23, *p* = 0.03; 95% CI = [−0.95, −0.06]), problem-solving (β = −0.21, *p* = 0.04, 95% CI = [−0.89, −0.03]), and personal–social (β = −0.22, *p* = 0.04; 95% CI = [−0.92,−0.03]) were predicted by having the father residing with their children. In turn, being a girl predicted a higher level of communication (β = 0.23, *p* = 0.03; 95% CI = [0.04, 0.85]), problem-solving (β = 0.20, *p* = 0.05; 95% CI = [0.00, 0.79]), and personal–social (β = 1.24, *p* = 0.01; 95% CI = [0.17, 1.01]) development.

Communication scores were higher in children who were attending preschool, but only when their mothers did not receive childcare support (unstandardized marginal B = 0.41 and 95% CI = [0.12, 0.70]). Thus, preschool attendance appears to improve children’s communication development Additionally, the interaction between mothers’ affection and mothers receiving support in the care of their children (β = −0.71, *p* = 0.06; 95% CI = [−0.82, 0.13]) predicted higher scores in problem-solving development. The interaction between the mother’s affection and the child’s gender (β = 1.24, *p* = 0.01; 95% CI = [0.17, 1.01]) showed higher personal–social scores.

It should be noted that mothers’ teaching scores (β = 0.32, *p* = 0.01; 95% CI = [0.09, 0.54]) and mothers dropping out of school (β = 0.25, *p* = 0.02; 95% CI = [0.13, 1.30]) were significantly associated with child problem-solving development only, but not with the other developmental areas.

Table 4 summarizes the regression model obtained based on a feasible solutions algorithm using ASQ-3′s communication, problem-solving, and personal–social development scores (n = 79) as the response. Categorical factors include the category to be compared to the reference category within parentheses.

## 4. Discussion

In the present study, we aimed to determine the relationship between some familial sociodemographic factors (e.g., a mother who had dropped out of school, employment status, support in the care of the child, child’s gender, fathers/grandparents living with their child, and children attending preschool) and parenting characteristics such as affection, responsiveness, encouragement, and teaching [2] of Chilean adolescent mothers, concerning communication, problem-solving, and personal–social developmental outcomes in typically developing children. Below we discuss how sociodemographic variables and adolescent motherhood are related to child development.

### 4.1. Sociodemographic Factors and Children’s Developmental Outcomes

Comparative analyses showed significant differences between child development based on certain sociodemographic variables. Children with employed mothers and mothers who had not dropped out of school had higher problem-solving developmental scores. Regarding the association of maternal employment and child cognitive development mentioned in the introduction, children of working Chilean mothers demonstrate more robust cognitive performance, given that maternal employment contributes to their parenting skills, leading to a higher level of child development [24,25]. The association between employment and parenting skills was also observed in this study since it was found that employed mothers had higher scores in the teaching domain; in turn, the teaching domain was significantly associated with child cognitive development. This result contributes to the discussion in the current literature on the reconciliation between work and child care [89], which has shown that the mother’s employment does not interfere with their children’s daily activities [90]. In fact, according to the findings of Huston and Rosenkrantz Aronson [91], mothers who worked for longer provided their children with higher-quality home environments and interaction. This issue will be addressed in future research, in which we intend to examine the association between maternal employment and parenting as factors influencing children’s developmental outcomes.

Regarding the difference in problem-solving development between children of mothers who had dropped out of school and those who had not, it is well known that socioeconomic status factors are significantly associated with cognitive development, making maternal education an important variable [92]. In the case of the sample in this study, the low level of education that adolescent mothers usually receive could be associated with disadvantaged social circumstances which put their children’s cognitive development at risk [19]. Mothers’ educational level, regardless of income, has been found to be a strong predictor of their children’s cognitive development [93]. In addition, higher educational levels in mothers who became pregnant during adolescence predicted better achievement of children in mathematics at the start of school [94]. Therefore, the relationship between adolescent mothers’ continuity of education and their children’s cognitive development is interesting since mothers who do not drop out of school can make a difference in their children’s cognitive development even when they belong to a low-income family. However, future studies with larger samples that include different levels of socioeconomic status are required to deepen our understanding of the relationship between school dropout by adolescent mothers and their children’s cognitive development.

We also found that children who resided with their fathers achieved significantly higher scores in communication, problem-solving, and personal–social development. Children living with both parents may receive more financial investment and spend more time with their fathers, which, in turn, would enhance child development [95]. Furthermore, the father’s presence can help in the child’s development through engagement in activities that contribute to their children’s learning and through the provision of emotional support and guidance. Indeed, the effectiveness of parenting depends on the quantity of time parents spend with their children and the quality of the relationships they establish with them [96]. This may explain why children who lived with their fathers achieved higher scores in developmental outcomes in our sample. Our results corroborate those of another study carried out on children of adolescent mothers, which found that children who were in contact with their fathers during the first eight years of life had higher levels of social–emotional functioning and reading than those who were not [47].

Another aspect of our results to consider is that mothers showed higher responsiveness, encouragement, and teaching scores when fathers cohabited. These maternal factors were related to their children’s development. In this context, the coresidence of mothers and fathers could be a protective factor for maternal well-being [97], and thus contribute to the quality of the dyadic interaction in their children’s development. More specifically, the absence of the father is an aspect that directly is related to stress in socioeconomically disadvantaged Chilean mothers [98]. In adolescent mothers, the father’s presence helps to reduce maternal depression [99]. In turn, a low level of mental health in adolescent mothers has been related to developmental problems in their children [30]. This factor was not considered in our study, but it is a relevant issue that could be considered in future studies on adolescent mothers.

Another sociodemographic variable associated with the children’s developmental outcomes was their gender. Specifically, girls scored higher than boys in communication, problem-solving, and personal–social development. Our results agree with studies that have shown that girls have higher levels of expressive language than boys. [100], and showed stronger social interaction during the preschool stage, because the forms of social and structured play appear earlier in girls, even though these differences vary over time [101]. Also, in more recent research on children aged from 4 to 7 years, girls showed higher cognitive ability and faster performance on mental tasks than boys [102]. So, it is reasonable to assume that the biological components that explain functional and morphological differences in the brain between boys and girls [103] lead to a higher level of performance in girls in all areas assessed.

The type of interactions that parents and children display may vary according to their gender and, in turn, be associated with their developmental outcomes. Adolescent mothers in our sample showed higher levels of responsiveness and teaching when interacting with girls than boys. Our findings are similar to previous research by Butler and Shalit-Naggar [104], who observed that mothers tend to exhibit higher levels of responsiveness towards girls than boys. Leaper [105] also reported that mothers show increased responsiveness and engage in more conversations with their daughters than with their sons. It is widely recognized that explanations and interactive conversations during play significantly contribute to a child’s development [106,107]. Therefore, both the biological factors that indicate that girls mature before boys and the parental characteristics of adolescent mothers when interacting with their children may explain why girls had higher scores in developmental outcomes than boys.

### 4.2. Parenting and Child Development

Parenting behaviors have been studied in various ethnic groups [108,109]. Recent literature has documented the association between parenting and child development, both in families with developmental delay disabilities [77,110,111] and those with typically developing children [112,113]. In line with other studies in which parenting was assessed with PICCOLO in adult mothers of typically developing children [114] and children with disabilities [77], the responsiveness domain showed the highest mean score, followed by affection, encouragement and finally teaching. Thus, it is possible to infer that the dyadic interaction between Chilean adolescent mothers and adult mothers with their children does not vary greatly regarding the frequency of parental behavior during free play or the association between the parenting domains and their children’s development.

Interestingly, the parenting domain scores in the current study sample were lower than in all previously mentioned studies. However, compared with the original study of parenting assessed using the PICCOLO in a low-income sample from the US that included European American, African American, and Latino families [108], in our sample, the mean scores of the affection and encouragement domains were lower, those of responsiveness were similar, and those of teaching were higher. Also, compared with a Turkish sample [109], all domain scores were lower with the exception of teaching.

Given the low economic and educational status of the mothers who participated in our study, it is surprising that the mean score for the adolescent mothers’ teaching domain was higher than in the other studies mentioned. Nevertheless, a study conducted with Chilean mothers aged from 15 to 44 years old found higher means in all domains than in our sample [21]. In future studies, it would be interesting to compare these parenting domains in Chilean adolescent and adult mothers to determine whether there are significant differences. This comparison is necessary because previous evidence has found differences in the support that adult and adolescent mothers receive [115]. In this context, adolescent mothers tend to be more depressive, feel less efficacy in their maternal role, and receive lower social support than adult mothers—all features that may affect their parenting [116].

The results of our study confirm the association between early parenting behaviors and the cognitive, linguistic, and social development of children of adolescent mothers. Our results corroborate those of previous research that has found a close relationship between adult mothers’ parenting and child development [117,118]. Specifically, our results show that responsiveness was associated with communication and problem-solving development. This is consistent with the findings of several previous studies of parenting interactions with children with typical development [45,48,59,60]. This result may represent a valuable contribution to the existing literature because previous work has found that maternal responsiveness in adolescent mothers acted as a mediator between socioeconomic risk and cognitive development [8].

In this study, we also found that children whose mothers displayed more encouraging behaviors during mother–child interactions showed higher scores in problem-solving development. Our results are consistent with studies that found that children of mothers who support their children’s autonomy (which is associated with the encouragement domain) achieved higher levels of executive function, which in turn is related to the children’s skills to solve problems [119].

Our findings also establish that adolescent mothers’ teaching is associated with communication, problem-solving, and personal–social development. The association between teaching and children’s linguistic, cognitive, and social development is an aspect clearly documented in previous literature [118,120,121], although few studies have analyzed this issue in adolescent mothers. However, previous studies of adolescent mothers have shown that having a poor language-learning home environment was related to lower language development in children [122].

This view is in line with the study by Shephard et al. [123], which established that due to factors such as living in impoverished environments and mental health problems, children born to adolescent mothers represent a population at high risk of impaired development. However, although this study focused on low-income adolescent mothers, risk variables such as maltreatment, abuse, or neglect, which are frequent factors in these vulnerable dyads [124], were not analyzed. Future studies of adolescent mothers should analyze the association between their context and parenting skills.

Given that adolescent motherhood is concentrated in low socioeconomic groups in Chile [125], there is a gap between low-income and high-income children’s development studies [126]. Our findings indicate that positive parenting in terms of responsiveness, encouragement, and teaching contributes to the cognitive, linguistic, and socio-individual development of children in our country. The significance of these results lies in the fact that the home environment and parenting competencies are significantly linked to children’s development [127]. From this perspective, it would be useful to design intervention programs to strengthen parental skills in low-income adolescent mothers and thus prevent cognitive and linguistic problems in their children, since these children are especially prone to suffer poor developmental outcomes [128].

In this study, the mother’s affection was not significantly related to the children’s cognitive, linguistic, or personal–social development. It has been demonstrated that maternal warmth moderates the impact of maternal intrusiveness on child anxiety and separation anxiety [129] and is associated with closeness and trust between mother and child. A study comparing adolescent and adult mothers interacting with their children found that adolescents spent more time in negative engagement with their children than their adult peers, producing insecure attachment [42]. Thus, maternal affection is more associated with secure attachment than children’s cognitive and linguistic development [130]. This may explain why adolescent mothers’ affection was not associated with any of the areas of child development evaluated.

### 4.3. Predictive Models for Child’s Development

Regarding the association of sociodemographic and parental factors with children’s development, the higher scores in communication development could be explained by higher maternal responsiveness, being a girl, having the father residing with their children, and the mother receiving childcare support (the last only when children did not attend preschool). Previous studies carried out on mothers and fathers in interactions with their children have concluded that affection is higher when parents interact with their daughters than with their sons [69,131,132]. Moreover, fathers were more attentively engaged with their daughters and used more analytical language than with their sons [133]. It is well known that parents’ responsiveness promotes expanded word learning in children’s early years [134]. This may partially explain why fathers residing with their children, the mother’s responsiveness, and the children’s gender all play an essential role in communication development. However, to determine whether this result is due to the evidence presented, it would be necessary to analyze the quality of the dyadic interaction between parents and their children. The complementary effect of the parenting competencies of fathers and mothers requires investigation, since it may explain the higher scores in communication development in girls than in boys. Nevertheless, this issue was not addressed in the current study. As we mentioned in the introduction, mothers who reside with the fathers of their children may suffer lower levels of distress [71], which, in turn, are associated with positive parenting [70]. Maternal health was not assessed in this research; this is a variable that should be considered in future studies examining factors related to parenting in adolescent mothers.

Regarding the lack of support that adolescent mothers receive in childcare, preschool centers contribute to reducing the gap between children from lower- and higher-income families. The preschool environment promotes linguistic and mathematical development by reducing problem behaviors caused by the rapid brain development experienced during the first years of life [135]. It seems reasonable to assume this association, even though in Chile the development of children aged from 6 months to 3 years who attend preschool centers and those who stay at home is similar. In our sample, children who attended preschool showed higher scores in communication, particularly if their mothers did not receive childcare support. This result suggests that for younger children, family variables may play a more important role than attending preschool [136]. However, when adolescent mothers do not receive support, the preschool center could be a good option. Preschool centers specialize in fostering strong relationships crucial for teaching and learning processes, thus improving children’s well-being and monitoring absences in children younger than 3 years due to illness during the winter, financial constraints hindering transportation, and family conflicts [137]. In any case, future studies need to analyze this interaction between the childcare support that adolescent mothers receive and their children’s attendance at preschool centers and its effects in later years because previous literature reports indicate that children who attend educational centers from an early age have a higher academic performance in the long term [138].

The problem-solving development score was higher for girls with higher mothers’ teaching scores, mothers who had not dropped out of school, and fathers residing with their children. Scientific evidence shows that the mothers’ educational level contributes to the cognitive development of their children, which would also be influenced by a cognitively stimulating home environment [127]. As mentioned above, our findings are consistent with recent research. Furthermore, when fathers engage positively in activities with their children, teaching activities can promote learning and development [139]. Future studies should try to determine the types of activities that the fathers of children of adolescent mothers perform, and whether they predict better development of children’s problem-solving skills.

Finally, children’s social–personal development scores were higher when fathers resided with them and in daughters of those mothers with higher levels of affection. Our findings are consistent with those of previous work reporting that fathers’ coresidence with the children of adolescent mothers during the first three years of life was related to lower externalization problems in children [71]. Also, the evidence has shown that maternal affection/warmth is related to stronger regulation and prosocial behavior in children [140,141] and is higher in daughters than in sons [142]. Although our sample did not show a direct correlation between a mother’s affection and children’s development, or any difference between boys and girls, we found that mothers show more affection towards girls when the fathers reside with them from an early age. More studies are required to analyze whether this phenomenon occurs in samples of adult and adolescent mothers and fathers.

### 4.4. Limitations and Future Directions for Research

This study adds to the previous scientific literature on positive parenting and its association with child development in adolescent mothers and typically developing children. However, we would like to point out some limitations. First, the study sample size might appear to be small since it only included 79 participants. Nevertheless, post hoc power analyses for this sample size indicate that, given a predictive capacity of R2 = 0.20 in models with five predictors and α = 0.05, a statistical power greater than 0.95 is still obtained. Thus, the sample size allowed us to reach adequate conclusions concerning the effects. Nevertheless, it is not possible to generalize these results since the sampling was intentional and its representativeness might thus be affected.

Additionally, it is necessary to remember that the presence of correlations does not imply causality. Specifically, in this descriptive and cross-sectional study, the term “predict” does not mean direct causality, but rather estimates child development scores based on scores of sociodemographic variables and parental variables, that is, the predictor variables. The fact that it was not a longitudinal study may be a limitation in so far as the quality of interaction may change depending on the age of the child [143], the age of the mother, and her level of education [21].

Another limitation to consider is that the tool we used to measure child development was a caregiver report rather than a behavioral observation measurement instrument of the kind applied in other research with typically developing children [144,145]. However, the ASQ-3 is a recommended and qualified screening tool for investigating children’s development [82], and UNICEF has recommended it for developing countries [146].

In addition, some internal consistency indices associated with PICCOLO subscales were low, in agreement with a study carried out by Rivero et al. [147] with a sample that included children with typical development. Although this might affect the psychometric properties of some of the tools, it is not an issue specific to the current study. In spite of this, PICCOLO has been applied to children in large samples in the United States in diverse ethnic groups [108], in Turkey [109], and in Spain [53]. These studies have demonstrated this instrument’s strong reliability and predictive validity for low-income families with diverse cultural backgrounds.

Lastly, the study did not include fathers. Our findings indicate that the levels of development of the children who lived with their fathers were higher in all the areas evaluated (communication, problem-solving, and personal–social). From this point of view, our results could have been more robust if we had analyzed the relationship between the parental dimensions of the fathers measured by PICCOLO and their children’s development, complementing this with the parental behaviors of the adolescent mothers, as has been conducted in other studies with adult parents [78,148]. In this study, few adolescent mothers lived with the father of their children, similar to other studies carried out in Chile [27]. More studies are needed to determine the consistency of our results.

## 5. Conclusions

The current study supports the idea that sociodemographic and parenting factors of adolescent mothers may explain their children’s developmental outcomes. The above findings will contribute to determining how the presence of the father and adolescent mothers’ responsiveness and teaching are associated with their children’s communication, problem-solving, and personal–social development. Adolescent mothers constitute a risk group in terms of social and economic factors and (frequently) their low educational level. Thus, promoting adolescent mothers’ knowledge of the stages of child development is an issue that should be considered when planning interventions to enhance parenting skills and child development in Chilean national programs. Our results highlight the need for social support for adolescent mothers, especially in relation to their parenting and support for continuity in their studies for better access to work, and promoting their emotional and family well-being.

## Figures and Tables

**Table 1 children-10-01778-t001:** Sociodemographic characteristics of adolescent mothers and their children.

Childcare Support	N	%
Childcare support provided	62	78.5
Childcare support not provided	17	21.5
Mother’s employment		
Employed	15	19.0
Unemployed	64	81.0
Dropped out school		
Yes	11	13.9
No	68	86.1
Child gender		
Male	42	53.2
Female	37	46.8
Preschool center		
Attends	28	35.4
Does not attend	51	64.6
Father co-habiting with mother and his child		
Yes	23	29.1
No	56	70.9
Grandparents co-habiting with mother and the child		
Mother, father, and children	6	7.6
Mother, grandparents, and children	56	70.9
Mother, grandparents, fathers, and children	17	21.5

**Table 2 children-10-01778-t002:** Descriptive summary of ASQ-3 Z and PICCOLO scores by sociodemographic factors.

	Communication	Problem-Solving	Personal–Social	Affection	Responsiveness	Encouragement	Teaching
	M ± SD (Min–Max)	M ± SD (Min–Max)	M ± SD (Min–Max)	M ± SD (Min–Max)	M ± SD (Min–Max)	M ± SD (Min–Max)	M ± SD (Min–Max)
Mother’s level of education							
Dropped out of school (n = 11)	0.03 ± 0.97 (−1.86−1.19)	−0.6 ± 0.88 (−1.88−1.23)	0.23 ± 1.01 (−1.51−1.66)	8.82 ± 2.23 (4.0−12.0)	9.64 ± 2.88 (3.0−13.0)	7.27 ± 2.97 (1.0−11.0)	8.82 ± 3.74 (2.0–14.0)
Not dropped out of school (n = 68)	−0.01 ± 0.96 (−2.06−2.09)	0.1 ± 0.94 (−2.34−1.7)	−0.04 ± 0.95 (−2.13−2.07)	10.09 ± 2.53 (3.0−14.0)	10.09 ± 2.50 (2.0−14.0)	9.03 ± 2.59 (2.0− 14.0)	8.56 ± 2.92 (0−14.0)
Statistical test	*t* = 0.12	*t* = −2.32 *	*t* = 0.84	*t* = −1.57	*t* = −0.55	*t* = −2.05 *	*t* = 0.26
(Effect size)	*d* = 0.04	*d* = 0.76	*d* = 0.28	*d* = 0.51	*d* = 0.18	*d* = 0.67	*d* = 0.09
Mother’s employment status							
Employee (n = 15)	−0.01 ± 0.79 (−1.27−2−09)	0.45 ± 0.81 (−0.83−1.38)	−0.05 ± 0.8 (−1.51−1.13)	10.40 ± 2,13 (7.0−13.0)	10.87 ± 1,685 (7.0 −13.0)	9.73 ± 1.944 (7.0 −14.0)	10.07 ± 2.086 (5.0–13.0)
Not an employee (n = 64)	0 ± 0.99 (−2.06−2−06)	−0.11 ± 0.96 (−2.34−1.7)	0.01 ± 0.99 (−2.13−2.07)	9.80 ± 2.60 (3.0−14.0)	9.83 ± 2.67 (2.0−14.0)	8.56 ± 2.81 (1.0− 14.0)	8.25 ± 3.12 (0−14.0)
Statistical test	*t* = −0.04	*t* = 2.07 *	*t* = −0.23	*t* = 0.84	*t* = 1.44	*t* = 1.53	*t* = 2.73 *
(Effect size)	*d* = 0.01	*d* = 0.60	*d* = 0.07	*d* = 0.24	*d* = 0.41	*d* = 0.44	*d* = 0.62
Support in the care of child							
Receiving support in the care of child (n = 62)	0.08 ± 0.94 (−2.06−2.06)	0.07 ± 0.96 (−1.93−1.7)	0 ± 0.98 (−2.13−2.07)	9.94 ± 2.40 (4.0−14.0)	10.18 ± 2.28 (3.0−14.0)	8.85 ± 2.64 (1.0−14.0)	8.65 ± 2.81 (2.0–14.0)
Not receiving support in the care of child (n = 17)	−0.3 ± 0.97 (−1.98−2.06)	−0.27 ± 0.89 (−2.34−1.04)	−0.01 ± 0.87 (−1.51−1.66)	9.82 ± 2.96 (3.0−14.0)	9.47 ± 3.34 (2.0−13.0)	8.53 ± 2.98 (2.0−12.0)	8.41 ± 3.81 (0−14.0)
Statistical test	*t* = 1.47	*t* = 1.31	*t* = 0.06	*t* = 0.16	*t* = 1.02	*t* = 0.44	*t* = 0.28
(Effect size)	*d* = 0.41	*d* =0.37	*d* = 0.02	*d* = 0.04	*d* = 0.28	*d* = 0.12	*d* = 0.08
Child gender							
Girl (n = 37)	0.25 ± 0.83 (−1.13−2.09)	0.25 ± 0.86 (−1.93−1.7)	0.29 ± 1.01 (−1.66−2.07)	10.05 ±2.33 (3.0−14.0)	10.68 ± 2.22 (2.0−14.0)	9.30 ± 2.33 (4.0−14.0)	9.24 ± 2.55 (2.0–14.0)
Boy (n = 42)	−0.22 ± 1.01 (−2.06−1.5)	−0.22 ± 0.99 (−2.34−1.7)	−0.26 ± 0.83 (−2.13−1.18)	9.79 ± 2.68 (4.0−14.0)	9.45 ± 2.68 (3.0−13.0)	8.33 ± 2.94 (1.0−14.0)	8.02 ± 3.31 (0−14.0)
Statistical test	*t* = −2.22 *	*t* = −2.21 *	*t* = −2.65 **	*t* = −0.47	*t* = −2.19 *	*t* = −1.60	*t* = −1.82 *
(Effect size)	*d* = 0.51	*d* = 0.50	*d* = 0.60	*d* = 0.11	*d* = 0.49	*d* = 0.36	*d* = 0.41
Preschool center							
Had attended preschool (n = 28)	−0.1 ± 1.07 (−2.06−2.06)	0.17 ± 0.96 (−1.93−1.38)	−0.13 ± 0.95 (−2.13−1.91)	10.04 ± 2.55 (5.0−14.0)	10.07± 2.36 (5.0 −13.0)	9.0 ± 2.66 (3.0−14.0)	8.79 ± 2.59 (3.0–14.0)
Had not attended preschool (n = 51)	0.05 ± 0.89 (−1.98−2.09)	−0.09 ± 0.95 (−2.34−1.7)	0.07 ± 0.96 (−1.7−2.07)	9.84 ± 2.52 (3.0−14.0)	10.0 ± 2.65 (2.0−14.0)	8.67 ± 2.74 (1.0−14.0)	8.49 ± 2.26 (0−14.0)
Statistical test	*t* = −0.65	*t* = 1.18	*t* = −0.91	*t* = 0.32	*t* = 0.12	*t* = 0.52	*t* = 0.41
(Effect size)	*d* = 0.16	*d* = 0.28	*d* = 0.22	*d* = 0.08	*d* = 0.03	*d* = 0.12	*d* = 0.10
Father							
Residing with his child (n = 24)	0.43 ± 0.93 (−1.86−2.06)	0.4 ± 0.7 (−0.83−1.7)	0.41 ± 0.82 (−1.12−1.59)	10.13 ± 2.68 (3.0−14.0)	10.88± 2.54 (2.0 −14.0)	9.63 ± 2.30 (4.0−12.0)	9.46 ± 3.16 (2.0–14.0)
Not residing with his child (n = 55)	−0.19 ± 0.91 (−2.06−2.09)	−0.17 ± 1 (−2.34−1.38)	−0.18 ± 0.96 (−2.13−2.07)	9.82 ± 2.46 (4.0−14.0)	9.65 ± 2.47 (3.0−14.0)	8.42 ± 2.79 (1.0−14.0)	8.22 ± 2.91 (0−13.0)
Statistical test	*t* = 2.77 **	*t* = 2.54 **	*t* = 2.61 **	*t* = 0.50	*t* = 2.0 *	*t* = 1.86 *	*t* = 1.70 *
(Effect size)	*d* = 0.69	*d* = 0.63	*d* = 0.65	*d* = 0.12	*d* = 0.49	*d* = 0.45	*d* = 0.42
Grandparents							
Mother, father, and children (n = 6)	0.13 ± 0.93 (−1.13−1.23)	0.40 ± 0.55 (−0.24−1.04)	0.22 ± 0.89 (−0.91−1.51)	10.0 ± 3.69 (3.0−14.0)	9.50 ± 4.14 (2.0−13.0)	9.83 ± 2.93 (4.0−12.0)	7.83 ± 4.31 (2.0−14.0)
Mother, grandparents, and children (n = 56)	−0.15 ± 0.93 (−2.06−2.09)	−0.16 ± 0.98 (−2.34−1.38)	−0.11 ± 0.99 (−2.13−2.07)	9.75 ± 2.53 (4.0−14.0)	9.77 ± 2.05 (3.0−13.0)	8.41 ± 2.83 (1.00−14.0)	8.18 ± 3.0 (0−13.0)
Mother, grandparents, father, and children (n = 17)	0.43 ± 0.95 (−1.87−2.06)	0.39 ± 0.87 (−1.53−1.70)	0.29 ± 0.81 (−1.12−1.59)	10.41 ± 2.03 (7.0−13.0)	11.06 ± 1.75 (8.0−14.0)	9.65 ± 1.90 (5.0−12.0)	10.24 ± 2.49 (3.0−14.0)
Statistical test	*F* = 2.54	*F* = 2.85	*F* = 1.31	*F* = 0.45	*F* = 1.87	*F* = 1.90	*F* = 3.42
(Effect size)	η^2^ = 0.06	η^2^ = 0.07	η^2^ = 0.03	η^2^ = 0.01	η^2^ = 0.05	η^2^ = 0.05	η^2^ = 0.08

Notes: * *p* < 0.05, ** *p* < 0.01.

**Table 3 children-10-01778-t003:** Pearson’s correlations between ASQ-3 Z scores and PICCOLO domain scores.

	2	3	4	5	6	7
1. Affection	0.57 **	0.74 **	0.33 **	0.00	0.10	0.06
2. Responsiveness		0.62 **	0.60 **	0.34 **	0.26 *	0.10
3. Encouragement			0.54 **	0.18	0.24 *	0.19
4. Teaching				0.27 *	0.35 **	0.24 *
5. Communication					0.36 **	0.35 **
6. Problem solving						0.37 *
7. Personal–social						

Notes: * *p* < 0.05 ** *p* < 0.01.

**Table 4 children-10-01778-t004:** Regression model of children’s developmental scores.

Child Development: Communication					
Variable	Estimate	SE	Standardized Beta	*t*	*p*
Intercept	−0.77	0.47		−1.63	0.11
Mother’s responsiveness	0.07	0.04	0.18	1.70	0.09
Gender (Female)	0.43	0.19	0.23	2.20	0.03 *
Father residing with his child (No)	−0.48	0.21	−0.23	−2.28	0.03 *
Receiving childcare support (Yes)	0.57	0.45	0.25	1.25	0.21
Child attending preschool (Yes)	0.53	0.22	0.27	2.46	0.02 *
Child attending preschool × Childcare support provided	−1.42	0.53	−0.55	−2.68	0.01 **
Adj. R^2^ = 0.26					
Child development: Problem-solving					
Variable	Estimate	SE	Standardized Beta	*t*	*p*
Intercept	−1.10	0.63		−1.74	0.09
Mother’s teaching	0.10	0.04	0.32	2.82	0.01 **
Mother dropped out of school (No)	0.68	0.28	0.25	2.45	0.02 *
Mother employed (No)	−0.33	0.24	−0.13	−1.35	0.18
Gender (Female)	0.38	0.19	0.20	1.98	0.05
Father residing with his child (No)	−0.44	0.21	−0.21	−2.15	0.04 *
Mother’s affection	0.01	0.05	0.03	0.27	0.79
Receiving childcare support (Yes)	1.25	0.86	0.54	1.44	0.15
Mother’s affection × childcare support provided	−0.16	0.08	−0.71	−1.88	0.06
Adj. R^2^ = 0.27					
Child development: Personal–Social	Estimate	SE	Standardized Beta	*t*	*p*
Variable					
Intercept	0.56	0.55		1.02	0.31
Mother’s teaching	0.06	0.04	0.19	1.69	0.09
Gender (Female)	−1.79	0.82	−0.94	−2.19	0.03 *
Father residing with his child (No)	−0.45	0.21	−0.22	−2.11	0.04 *
Mother’s affection	−0.10	0.05	−0.26	−1.87	0.07
Mother’s affection × child’s gender	0.22	0.08	1.24	2.83	0.01 **
Adj. R^2^ = 0.20					

Notes: * *p* < 0.05, ** *p* ≤ 0.01.

## Data Availability

The data presented in this study are available on request from the corresponding author. The data are not publicly available due to the data are part of an ongoing research project. For this reason, an anonymized version of the database was created that includes the relevant variables to reproduce the analyzes if required.

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
