# Peer review of "Positive Parenting and Sociodemographic Factors Related to the Development of Chilean Children Born to Adolescent Mothers"

_children, 2023, doi:10.3390/children10111778_

Round 1

Reviewer 1 Report

Comments and Suggestions for Authors

The study submitted for review aimed to investigate the relationship between demographic variables in the family context and parenting behaviour among adolescent mothers. The study is complete and the results presented in it are complete. The discussion is conducted fairly and points in the right directions for interpreting the results obtained. 
Nevertheless, the paper contains many editorial errors. Some of those found I present below, but I feel that the work needs a very important review to catch all of them. 

1/ The authors use literature references in the middle of sentences:

the emotional support given to their children being significant factors [17]. Given the so- (line: 53) cioeconomic challenges faced by adolescent mothers in Latin America, [18] emphasized (54) 
the importance of incorporating an examination of co-parenting relationships, particu- (55) 

obío region, known for having some of the highest poverty rates in Chile [21], also reports (61)

Moreover, it has been demonstrated that mothers are more affectionate with girls [82] (201) 
and when their distress levels are lower [83]. Lower maternal distress is associated with (202)

of Market Researchers 2019 [87] and children of typical development. Within this group (235)

that included European Americans, African Americans, and Latino American families (549) 
[115], in our sample, the means of the affection and encouragement domains were lower, (550) 
responsiveness was similar, and teaching was higher. Also, compared with a Turkish sam- (551) 
ple [116], all domains were lower, except teaching. (552) 

2/ Unnecessary characters such as ")" appear:

been carried out with adult mothers [23-26] (. Meanwhile, few studies have explored the (67)

ment across the children's gender [81]). (200)

variables, such as parent's age, employment, and educational level [52] (), it is crucial to (142)

3/ There is no consistent identification system and there are no dots before numerical values

reliability (Cronbach Alpha) for the total instrument was 88 (.76 for affection, .85 for re- (297) 

Chilean adolescent mothers (n = 79), the reliability (Cronbach α) for the total instrument 

4/There are different styles of citation

development [100] (Schady, 2011) and that higher educational levels in mothers who be- (476) 

5/ The authors used an impact term that could suggest that, from the study in question, an impact could be concluded ... this should be changed 

siveness and teaching influence their children's communication, problem-solving, and (724 )

Especially as they themselves point out that research in this direction should be expanded in the future: 

Although it has been shown that parenting depends on various sociodemographic 141 
variables, such as parent's age, employment, and educational level [52] (), it is crucial to 142
 investigate how these factors influence the development of children of adolescent mothers
.

Reviewer 2 Report

Comments and Suggestions for Authors

Dear Authors, This is a very important area of research and it is clear from the manuscript that you are most concerned to help the young women and their children.  The manuscript needs to be revised, the introduction is too long and the text needs to be reorganised.  the novel element of the study must be made clear. Inappropriate use of linkers and poor paragraph structure undermines the quality of the text, so this should be addressed, preferably by a native English speaker who understands your research area.  Please see the detailed comments on the manuscript attached.  With very best wishes

Comments on the Quality of English Language

The English is average quality. Inappropriate use of linkers and poor paragraph structure undermines the quality of the text, so this should be addressed, preferably by a native English speaker or a professional editing service that fully understands the subject area. 

Round 2

Reviewer 2 Report

Comments and Suggestions for Authors

Thank you for addressing the points raised.  Best wishes

Comments on the Quality of English Language

The normal MDPI editing process should pick up any other small errors.